# Experimental Cyclic Heat Stress on Intestinal Permeability, Bone Mineralization, Leukocyte Proportions and Meat Quality in Broiler Chickens

**DOI:** 10.3390/ani12101273

**Published:** 2022-05-16

**Authors:** Alessandro Rocchi, Jared Ruff, Clay J. Maynard, Aaron J. Forga, Roberto Señas-Cuesta, Elizabeth S. Greene, Juan D. Latorre, Christine N. Vuong, Brittany D. Graham, Xochitl Hernandez-Velasco, Guillermo Tellez, Victor M. Petrone-Garcia, Lauren Laverty, Billy M. Hargis, Gisela F. Erf, Casey M. Owens, Guillermo Tellez-Isaias

**Affiliations:** 1Department of Poultry Science, University of Arkansas, Fayetteville, AR 72701, USA; ajrocchi@uark.edu (A.R.); jaruff@uark.edu (J.R.); cjm019@uark.edu (C.J.M.); ajforga@uark.edu (A.J.F.); rsenascu@uark.edu (R.S.-C.); esgreene@uark.edu (E.S.G.); jl115@uark.edu (J.D.L.); vuong@uark.edu (C.N.V.); bmahaffe@email.uark.edu (B.D.G.); memotellez98@gmail.com (G.T.J.); lmlavert@uark.edu (L.L.); bhargis@uark.edu (B.M.H.); gferf@uark.edu (G.F.E.); cmowens@uark.edu (C.M.O.); 2Departamento de Medicina y Zootecnia de Aves, Facultad de Medicina Veterinaria y Zootecnia, Universidad Nacional Autonoma de Mexico, Mexico City 04510, Mexico; xochitlh@fmvz.unam.mx; 3Facultad de Estudios Superiores Cuautitlan, Universidad Nacional Autonoma de Mexico, Cuautitlan Izcalli 54121, Mexico; petrone@unam.mx

**Keywords:** bone mineralization, chickens, enteric inflammation, heat stress, parts weight

## Abstract

**Simple Summary:**

Recently, our laboratory published a heat stress model to induce gastrointestinal leakage in broiler chickens. Nevertheless, the model describes continuous heat stress (HS) conditions from day 21 until 42 days of age. The current brief report intends to present an alternative model (12 h/day), particularly more in accordance with real-world conditions, to evaluate nutraceuticals or products that can reduce the severity of HS under experimental conditions. Results of this study confirm what has been extensively reported by numerous scientists. Cyclic HS increased leaky gut, reduced performance, bone mineralization, and meat quality in broiler chickens.

**Abstract:**

The goal of this research was to assess cyclic heat stress on gut permeability, bone mineralization, and meat quality in chickens. Two separate trials were directed. 320 day-of-hatch Cobb 500 male chicks were randomly assigned to four thermoneutral (TN) and four cyclic heat stress (HS) chambers with two pens each, providing eight replicates per treatment in each trial (*n* = 20 chicks/replicate). Environmental conditions in the TN group were established to simulate commercial production settings. Heat stress chickens were exposed to cyclic HS at 35 °C for 12 h/day from days 7–42. Performance parameters, intestinal permeability, bone parameters, meat quality, and leukocyte proportions were estimated. There was a significant (*p* < 0.05) reduction in body weight (BW), BW gain, and feed intake, but the feed conversion ratio increased in chickens under cyclic HS. Moreover, HS chickens had a significantly higher gut permeability, monocyte and basophil levels, but less bone mineralization than TN chickens. Nevertheless, the TN group had significant increases in breast yield, woody breast, and white striping in breast fillets compared to HS. These results present an alternative model to our previously published continuous HS model to better reflect commercial conditions to evaluate commercially available nutraceuticals or products with claims of reducing the severity of heat stress.

## 1. Introduction

Temperatures have progressively increased in intensity and duration in recent years [1], and the frequency and severity of heatwaves are predicted to continue to rise in the future [2]. This situation is critical for the poultry industry because poultry is particularly vulnerable to heat stress (HS) due to the absence of sweat glands and increased metabolism [3].

Poultry, in general, are particularly susceptible to heat stress. Birds have developed various techniques to maintain physiological homeostasis and regulate core temperature to compensate for the lack of sweat glands, including convection, evaporation, and radiation [4]. However, during heat stress, the animal absorbs more heat than is dissipated [5]. Because of the demands on development rate and feed efficiency, there is no corresponding improvement in cardiovascular and respiratory functionality in modern broiler chickens [6,7]. In chickens, HS can manifest in a variety of ways, from mild discomfort and illness to extensive organ damage and death [8]. Several investigators have shown the detrimental effects of HS on performance parameters [5,9,10], meat quality [11], and intestinal permeability [9,10,12,13,14]. At the moment of writing this brief report, typing the keywords “cyclic heat stress in chickens” in Google Scholar, the result shows 23,300 scientific manuscripts published.

Chronic stress and persistent inflammation are detrimental to modern animal production operations. Any cause of long-term stress, no matter where it comes from, will induce oxidative stress and, if persistent, chronic intestinal and systemic inflammation [15,16,17]. Chronic inflammation has been linked to decreased bone density in epidemiological, clinical, and animal research [18,19]. In broilers, recent studies have shown that HS increases inflammation and bone mineralization [20,21,22], and similar results have been confirmed by our laboratory mineralization [14,23].

Given the increase in antibiotic-resistant zoonotic bacterial illnesses, there is a growing public and scientific interest in therapeutic and subtherapeutic antimicrobials administered to animals [24]. Social pressures led to curbs on antibiotic usage in poultry and livestock enterprises. Substitutes to antibiotic growth promoters (AGP) are needed in modern livestock farming. Recent research suggests that nutritional therapies to stress, disease, and chronic inflammation may be helpful as alternatives to antibiotics in some circumstances [25,26]. Disease resistance improvement in non-antibiotic raised animals has been shown to benefit animal health, welfare, and production, as well as the microbiological safety of animal products. Pressure from consumers to remove growth-promoting antibiotics and restrict therapeutic usage of existing medicines has led to new feed additives. Reliable experimental models such as cyclic heat stress could help to evaluate nutraceuticals as alternatives to AGP such as probiotics, prebiotics, organic acids, phytochemicals, and trace minerals that have shown promising results to improve the intestinal microbial balance, metabolism, and gut integrity by several investigators.

Recently, our laboratory published a heat stress model to induce gastrointestinal leakage in broiler chickens [9]. Nevertheless, the model describes continuous heat stress conditions from day 21 until 42 days of age. The current paper intends to present an alternative model, particularly more in accordance with real-world conditions, to evaluate nutraceuticals or products that can reduce the severity of heat stress under experimental conditions.

## 2. Materials and Methods

### 2.1. Facilities and Experimental Design

Two independent trials were run in environmentally controlled rooms equipped with air conditioning units and a digital thermostat. Trial 1 was conducted in the spring (March–May), and Trial 2 was conducted in the fall (November–December). In each trial, day-of-hatch Cobb 500 male chicks (320 total) were assigned at random to four thermoneutral (TN) and four cyclic heat stress chambers (HS) with 8 replicates per treatment, and 20 birds per pen, with its own feeder and watering system. Diets utilized in this study were prepared to give adequate nutrients and were adjusted to breeder recommendations [27] without growth promoters. In both trials, the environment for the TN group was established to simulate commercial production settings (temperature) for the first 21 d in all chambers by imposing a gradual reduction on temperature from 32 °C to 24 °C with relative humidity at 55 ± 5%. Temperature 24 °C and humidity 55 ± 5% were maintained from d 21–42. Lighting was provided constantly (24 h) during both trials in both groups. For the cyclic HS group, the environment was established to simulate commercial production settings (temperature, light) for the first six days. From day 7 to day 42, HS chickens were exposed to cyclic HS at 35 °C for 12 h/day. During the non-HS times of the day, HS chickens were kept at the same temperature settings as the TN chickens. The relative humidity stayed unchanged at 55%. On day 18 of age, eight chickens per replicate were chosen at random to have a Thermochron temperature logger implanted orally (iButton, DS1922L, Embedded Data Systems, Lawrenceburg, KY, USA). Loggers remained in the gizzard to measure the temperature every minute for the first two hours, then every hour thereafter [28]. At d 10, 28, and 42, body weight (BW), body weight gain (BWG), feed intake (FI), and feed conversion ratio (FCR) were evaluated on a replicate basis. On days 21 and 42, three randomly selected chickens per replicate (*n* = 24) were orally gavaged with 8.32 mg/kg body weight fluorescein isothiocyanate-dextran (FITC-d, MW 3–5 KDa; Sigma-Aldrich Co). Birds were euthanized by CO_2_ inhalation one hour after FITC-d administration. Serum FITC-d levels were used to assess intestinal permeability as previously published [29]. Tibias were removed to evaluate bone mineralization in accordance with previously stated [30].

In Exp 2 only, heparinized blood (1 mL) was collected from the wing vein of one broiler per replicate (*n* = 8) on day forty-two. Blood was used to prepare Wright-stained blood smears to examine the proportions (%) among white blood cells (WBC) [31]. The WBC concentration was determined by an automated hematology analyzer (Cell-Dyn; Abbot Diagnostics, Abbott Park, IL, USA). Additionally, eight birds were picked for each replicate (*n* = 64) to examine carcass yield and meat quality, as described below.

The Institutional Animal Care and Use Committee at the University of Arkansas followed protocol number 21,018 for animal handling procedures.

### 2.2. Processing Parameters, Carcass, and Meat Quality Parameters

At the University of Arkansas Pilot Processing Plant, sixty-four broilers from each group (*n* = 8 chickens/replicate) were processed using in-line commercial equipment according to the practices of Mehaffey et al. (2006) [32]. Directly after deboning, the degree of hardness for woody breast (WB) and white striping (WS) of whole breast fillets were determined using a tactile evaluation scale for WB [33]. Fillets were also scored for visual assessment of WS as previously described [34,35]. For classification purposes, each breast myopathy was scored in 0.5 increments. To eliminate variability, one trained person rated all fillets for myopathies. Each butterfly fillet was halved into left and right. The left fillets were used for measuring pH and color, whereas texture analysis was conducted on the right fillets. The removed weights were recorded for drip loss analysis. A probe was used to measure muscle pH (Model Testo 205, Testo Inc., Sparta, NJ, USA). The color values of lightness (L*), redness (a*), and yellowness (b*) were calculated as an average from three distinct places with a Minolta colorimeter (CR-400, Konica Minolta, Ramsey, NJ, USA). The color was recorded under a D65 illuminant utilizing a 2-degree observer in accordance with the American Meat Science Association guidelines [36]. The fillet length, width, cranial thickness (H1), and caudal thickness (H2) were measured using calipers [32]. These were vacuum sealed and maintained at −20°C till cook loss and Meullenet-Owens Razor Shear (MORS) were determined. The fillets were defrosted at 4 °C for 24 h before cooking. Each fillet was cooked separately in an air convection oven (Model E101-E, Duke Manufacturing Company, St. Louis, MI, USA), cooked on high wire racks in covered aluminum-lined pans to 76 °C internal end-point. The cook loss was determined as a percentage of the starting weight differential between the fillets. For texture study, the cooked fillets were individually wrapped in aluminum foil and refrigerated overnight at 4 °C. Objective tenderness was evaluated using the MORS technique [37,38]. The MORS force (N; MORSF), MORS energy (N mm; MORSE), and sample peak counts (MORS-PC) were used to predict meat softness.

### 2.3. Compression Force

Right halve fillets were then used for drip loss determination and compression analysis [37]. Drip loss was determined using the % loss method, which involved comparing deboned weight to postmortem weight 24 h later. Individual fillets were crushed to 20% of their height three times in different cranial regions using a 6-mm flat probe on a texture analyzer with processing data software. (Model TA-XT2 Plus, Texture Technologies Scarsdale, NY/Stable Micro Systems, Godalming, Surrey, UK). The trigger force was set to 5 g, the probe height was set to 55 mm, the pre and post-probe speeds were both set to 10 mm/s, and the probe test speed was set to 5 mm/s.

### 2.4. Statistical Analysis

All data were subjected to Student’s *t*-test in a completely randomized design using SAS software, where statistically significant differences between the means were set at a *p*-value of less than 0.05 [39]. The chickens’ body temperature was logged (*n* = 1250) from eight chickens per replicate. For performance parameters, the experimental unit was each replicate pen (*n* = 8 with 20 chickens/pen). For serum and bone parameters, the experimental unit was *n* = 24 (3 chickens/replicate). For the proportions and concentrations of white blood cells, one broiler per replicate (*n* = 8) was the experimental unit. For carcass and part yields, *n* = 64 was the experimental unit.

## 3. Results

The results of the evaluation of body core temperature, BW, BWG, FI, FCR, serum FITC-d, and bone parameters in broiler chickens during cyclic heat stress in trial 1 and trial 2 are summarized in Table 1. A significant increase in average body core temperature during the trial was observed in the HS group compared with the TN group (Table 1).

There were no statistically significant differences in both trials on any of the performance variables at day ten between TN or HS broilers. However, in both independent trials, a substantial reduction in BW, BWG, and FI was discerned in chickens that were subjected to cyclic HS compared to TN chickens at 28 and 42 days. FCR was not affected during the grower phase between TN or HS groups in both experiments. Nevertheless, an important worsening in FCR was noticed in TN compared with HS group at 42 days (Table 1).

In both trials, a significantly lower serum concentration of FITC-d was observed at 21 and 42 days in TN chickens compared with HS chickens (Table 1).

Tibia strength was not affected at 21 days in both experimental groups and studies. However, tibia strength was significantly higher in TN chickens at day 42 compared with the HS group in both trials (Table 1). Similar results were observed in total ash from the right tibia in both independent trials (Table 1).

The proportions (%) of monocytes and basophils were higher in 42-day-old broilers reared under HS (5.68 ± 0.69 ^a^) compared to TN (2.35 ± 0.31 ^b^) conditions. No significant differences were observed in the proportions (%) for lymphocytes, heterophils, or eosinophils between both groups in trial 2 (data not shown).

The evaluation results of the whole carcass and carcass parts yield (%), muscle myopathy scores, and meat quality attributes of broilers from TN and HS groups in Trial 2 are summarized in Table 2. The TN group showed a significantly higher breast yield compared to HS chickens; however, wing yield and whole leg yield were significantly lower than in HS chickens (Table 2). Interestingly, these findings also were associated with a significantly higher incidence of WB, WS, and a* value in the breast fillets of TN broilers compared to HS broilers, but significantly lower MORS force, energy, and peak counts (Table 2).

## 4. Discussion

In the present study, HS had a detrimental effect in performance parameters, carcass weight, and breast yield at processing, as has been shown previously by several investigators [5,9,10,11]. These changes were associated with lower myopathies, drip loss, compression force, and a* value. As expected, cooked meat quality attributes (cook loss, MORS force, MORS energy, and MORS peak counts) were altered by HS exposure. Interestingly, increasing breast size increased the overall occurrence of muscle myopathies in the TN group. The detrimental effects of HS and meat quality in broiler chickens are extensively reviewed by 6060 scientific manuscripts (Google Scholar keywords “myopathies and heat stress and chickens”); hence, the discussion of these effects is out of the objectives of the present brief report.

Chickens exposed to 12 h of cyclic HS for five consecutive weeks severely affected performance and intestinal permeability, as was demonstrated by a substantial rise in serum FITC-d levels. Additionally, the increased proportions and concentrations of circulating monocytes, and proportion of circulating basophils observed in HS chickens, also is in line with systemic inflammatory activity in response to microbes and/or microbial components (e.g., lipopolysaccharide) that crossed the HS-associated leaky gut barrier [40,41]. Increased serum FITC-d levels have been observed during continuous HS conditions in broiler chickens [9,14]. The intestinal epithelium has two crucial functions: preventing the admission of harmful intraluminal bacteria, allergens, and toxins, and allowing the selective transfer of food nutrients and electrolytes into circulation. Alterations in gut permeability are connected with bacterial translocation in the portal and/or systemic circulation in several types of leaky gut syndromes leading to systemic bacterial infections and chronic systemic inflammation, as has been shown by previous studies conducted in our laboratory [40,42,43]. As heat stress reduces the expression of tight junction proteins, germs and antigens escape into the circulation, causing severe chronic inflammation [13]. Chronic HS has been demonstrated to cause broilers to reduce performance [5,11]. HS can cause permeability in intestinal epithelial tissues by modifying the synthesis and distribution of tight junction proteins [10,41].

Chronic inflammation in the intestine is thought to be the root cause of 90% of all diseases [44]. Moreover, the composition of the gut microbiota is greatly affected by the diet ingredients as well as the immune and neuroendocrine systems [45,46]. The neuroendocrine network that connects the brain, the enteric nervous system, intestinal microbiota, and the gut-associated lymphoid tissue profoundly impacts the delicate gut barrier [47,48,49]. As a result, gut integrity is crucial in maintaining a good balance between health and disease [50,51]. In the GIT mucosa, dysbiosis (loss of equilibrium of the microbiota) leads to loss of intestinal integrity, increase intestinal permeability, and bacterial translocation [16,52].

In both trials, a significant increase in intestinal permeability was linked to decreased bone strength at day 42. These findings, too, are consistent with other studies (both in mammals and chickens) indicating that inflammation affects bone mass and strength while also interfering with bone healing and regeneration [18,19,20,21,22,53].

The chicken has served as an important experimental system for developmental biology, immunity, and microbiology for over two millennia [54,55]. This has resulted in numerous key discoveries in these fields [56]. The findings of this study confirm that it is possible to utilize cyclic heat stress as a model to evaluate commercially available nutraceuticals or products with claims of reducing the severity and adverse effects of heat stress by their anti-inflammatory, anti-oxidant, and immune-modulatory properties.

## Figures and Tables

**Table 1 animals-12-01273-t001:** Evaluation of body core temperature, body weight (BW), body weight gain (BWG), feed intake (FI), feed conversion ratio (FCR), serum FITC-d, and bone parameters in broiler chickens during cyclic heat stress in trials 1 and trial 2.

	Trial 1		Trial 2	
Variable	Thermoneutral	Heat Stress	Thermoneutral	Heat Stress
**Body core temperature (°C)**	40.17 ± 0.87 ^b^	42.37 ± 0.65 ^a^	40.68 ± 0.94 ^b^	42.11 ± 0.30 ^a^
**Days (11–28)**				
BW (g)	1674.4 ± 29.28 ^a^	1437.3 ± 26.89 ^b^	1574.4 ± 29.28 ^a^	1337.3 ± 26.89 ^b^
BWG (g)	1432.6 ± 24.32 ^a^	1205.3 ± 23.54 ^b^	1332.6 ± 24.32 ^a^	1105.3 ± 23.54 ^b^
FI (g)	2387.8 ± 35.75 ^a^	1999.5 ± 24.46 ^b^	2287.8 ± 35.75 ^a^	1947.5 ± 24.46 ^b^
FCR	1.444 ± 0.014	1.469 ± 0.019	1.454 ± 0.014	1.459 ± 0.019
**Days (0–42)**				
BW (g)	3035.8 ± 78.44 ^a^	2229.3 ± 52.55 ^b^	3207.3 ± 82.77 ^a^	2585.7 ± 17.16 ^b^
BWG (g)	2982.1 ± 78.01 ^a^	2172.3 ± 51.94 ^b^	1626.3 ± 92.62 ^a^	1263.5 ± 26.13 ^b^
FI (g)	4961.4 ± 129.66 ^a^	3968.3 ± 123.51 ^b^	5125.0 ± 80.81 ^a^	4379.3 ± 95.17 ^b^
FCR	1.635 ± 0.013 ^b^	1.781 ± 0.038 ^a^	1.581 ± 0.030 ^b^	1.695 ± 0.045 ^a^
**Serum FITC-d (ng/mL)**				
Day 21	15.85 ± 7.10 ^b^	48.84 ± 10.17 ^a^	44.29 ± 14.31 ^b^	141.73 ± 15.74 ^a^
Day 42	87.00 ± 17.47 ^b^	177.05 ± 15.73 ^a^	28.13 ± 10.55 ^b^	128.61 ± 10.73 ^a^
**Tibia break strength (kg)**				
Day 21	18.93 ± 0.86	16.79 ± 0.94	16.50 ± 0.47	16.12 ± 0.46
Day 42	39.71 ± 2.54 ^a^	22.37 ± 1.37 ^b^	34.18 ± 0.49 ^a^	24.43 ± 0.43 ^b^
**Total ash from tibia (%)**				
Day 21	55.07 ± 0.56	52.57 ± 0.86	51.62 ± 0.32	51.43 ± 0.28
Day 42	57.78 ± 0.66 ^a^	51.33 ± 0.76 ^b^	52.25 ± 0.44 ^a^	41.87 ± 0.59 ^b^

Data expressed as mean ± SE. ^a,b^ Values within rows with different superscripts differ significantly per experiment (*p* < 0.05). For performance, eight replicates per treatment group (*n* = 20 chickens/replicate). On days 21 and 42, blood and tibias were collected from three randomly selected chickens per pen (*n* = 24 per treatment group).

**Table 2 animals-12-01273-t002:** Carcass and part yields (%), muscle myopathy scores, and meat quality attributes of broilers from thermoneutral and cyclic heat stress groups in trial 2.

Variable	Thermoneutral	Heat Stress
**Carcass and parts yields (%)**		
Live weights (g)	3282 ± 46.54 ^a^	2715 ± 37.55 ^b^
Hot carcass	72.16 ± 0.52	71.98 ± 0.43
Fat	1.40 ± 0.03	1.41 ± 0.02
Chilled carcass	74.37 ± 0.64	74.27 ± 0.52
Wing	7.67 ± 0.05 ^b^	8.00 ± 0.02 ^a^
Breast	20.17 ± 0.65 ^a^	18.49 ± 0.42 ^b^
Tender	3.55 ± 0.04	3.49 ± 0.03
Whole leg	22.51 ± 0.24 ^b^	23.55 ± 0.32 ^a^
Rack	20.47 ± 0.64	20.73 ± 0.52
**Muscle myopathy scores and** **raw meat quality attributes**		
Woody breast	0.63 ± 0.005 ^a^	0.41± 0.002 ^b^
White striping	0.87 ± 0.002 ^a^	0.78± 0.003 ^b^
Drip Loss (%) ^1^	1.54 ± 0.03 ^a^	1.27 ± 0.02 ^b^
pH	5.85 ± 0.04	5.87 ± 0.03
Lightness (L*)	55.83 ± 0.42	55.63 ± 0.52
Redness (a*)	3.21 ± 0.01 ^a^	2.82 ± 0.03 ^b^
Yellowness (b*)	8.71 ± 0.04	8.48 ± 0.02
**Cooked meat quality attributes**		
Compression force (N)	5.20 ± 0.04 ^b^	8.01 ± 0.05 ^a^
Cook loss (%) ^1^	29.12 ± 0.44 ^b^	31.49 ± 0.53 ^a^
MORS ^2^ Force (N)	13.73 ± 0.32 ^b^	14.67 ± 0.37 ^a^
MORS ^2^ Energy (N.mm)	184.66 ± 1.42 ^b^	196.93 ± 2.27 ^a^
MORS ^2^ PC ^3^	9.13 ± 0.03 ^b^	10.81 ± 0.04 ^a^

^a,b^ Means without a common superscript were determined to be significantly different (*p* < 0.05). All values reported are on a percent basis in relation to live weight (*n* = 64 per treatment group). ^1^ Drip loss and cook loss were calculated as the percent loss method of initial minus final weight. ^2^ MORS—Meullenet-Owens Razor Shear. ^3^ PC—Peak counts.

## Data Availability

Not applicable.

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
