# Peer review of "Experimental Cyclic Heat Stress on Intestinal Permeability, Bone Mineralization, Leukocyte Proportions and Meat Quality in Broiler Chickens"

_animals, 2022, doi:10.3390/ani12101273_

Round 1
Reviewer 1 Report
This study measured a variety of parameters and conducted two trials to investigate cyclic heat stress on broilers. It provides more information for the industry to develop strategies to alleviate the effects of heat stress on broilers.
However, I think authors should work on how they deliver the information to readers. The Introduction was too general without focusing on heat stress and broilers. In addition, the motivation and significance of the current study are unclear. After reading the Introduction, I don’t know why the authors need to conduct this study, what the difference is between this study and any other previous studies, what research questions can be answered by this study, and why the authors chose those parameters to measure and investigate.
Moreover, the authors conducted two trials without comparing the results between them or discussing them. Then, I don’t understand the point of these two trials in the study.
Additionally, I strongly recommend authors rewrite the entire Discussion and Conclusions sections. There are many interesting findings that were not discussed at all. Currently, it seems that the Discussion consists of one paragraph of Introduction followed by one paragraph of results interpretation without adequate and appropriate discussion.
Importantly, authors should pay attention to choosing the right words and the way to structure the sentence to make the manuscript scientific sound. Other than that, many grammar issues throughout the manuscript reduce the interest in reading. Please see my specific comments below:
Line 19-22: Please consider rephrasing this sentence. It is too long and hard to follow.
Line 29: It’s better to use “these results” rather than “these data.”
Line 30: validate
Line 33: white blood cell counts?
Line 36: In this part, the rearing conditions for the TN group should also be described.
Line 40: You don’t start a sentence with an abbreviation or number. Please correct it throughout the manuscript.
Line 52-53: This sentence is incomplete or incorrect. Effects of HS range from?
Line 58: Authors should pay attention to the words they choose in the manuscript. Because “shut down” is not scientifically sound.
Line 55-68: This whole paragraph introduces various physiological responses to stress. However, “stress’ is a vast concept that covers various types of stressful situations, like social stress, disease stress, etc. This study focuses on heat stress. I suggest the authors rephrase this paragraph and focus more on chickens’ responses to heat stress, particularly cyclic heat stress.
Line69-70: Where is the citation for epidemiological, clinical research? It’s better to put the citation right behind instead of at the end of the sentence.
Line 69-74: In this part, it is not clear why this study is needed, how this research is different from previous studies, or what research gap can be filled by the current research? The first two paragraphs of the Introduction are too general. Then the third paragraph suddenly jumped into heat stress on broilers. Authors should work on improving the transitions and content of the Introduction.
Line 78-79: I’m confused by the experiment set-up regarding these two trials. If animals were kept in chambers where the temperature was controlled, then why did the authors repeat the same trial twice in two seasons/periods. But authors did not statistically compare the traits measured during the two trials if they wanted to investigate the season/period effects. Please clarify why two trials are needed here and discuss the differences in the results between the two trials in the Discussion.
Line 97-99: It’s not clear that BW and BWG were measured on individual or pen level? Were all the individuals in each pen or all the pens/replicates measured? Same for FI and FCR, please clarify.
Line 99: Per replicate or per pen? As I understand they are the same, but please choose one.
Line 122: You mean “Trial 2” instead of “Exp 2”? Please choose one and keep consistency throughout the manuscript. It isn’t very clear because it could be exp1 and exp2 within each trial.
Line 168: “were statistically different.” Please check the grammar of this sentence.
Line 169-170: This information should also be mentioned around Line 97-99.
Line 167-173: I’m curious that why the authors did not compare the measures between the two trials. Authors may consider trials as replicates and include this factor in the model when analyzing those traits measured during both trials.
Line 174: I suggest adding a table that summarizes the comprehensive descriptive statistics of all the measures considered in this study.
Line 195: Table 2, results in the last four rows are for D0 to D42 or D28 to D42? I also noticed that chickens in Trial 2 had smaller BW, BWG, and FI compared to chickens in Trial 1 during D0 to 28. I was curious if the authors compared these measures between two trials and if they have any thoughts on how to explain these differences between the two trials.
Line 247-255: This paragraph contains repeated information as in the Introduction section. I suggest moving this part to the Introduction section.
Line 260-262: If authors found something interesting, they should discuss it more instead of just repeating the results in the Discussion section.
Line 269-272: This paragraph seems to come from nowhere. It could be a part of the Introduction section. If the authors wanted to put it in the Discussion, they have to better connect it with their findings (the following paragraph). This paragraph lost the connection and did not flow well.
Line 277-278: Please add a citation for this sentence for the microbiome part.
Line 280: Discuss more of these interesting results.
Line 283-296: Same here. This paragraph started by introducing the effects of HS on the intestines, which has no connection to what the authors found in this study. Authors should not write a paragraph Introduction and follow with a paragraph repeating the results in the Discussion!
Line 315-319: It is NOT an appropriate conclusion for a scientific paper. The authors did not summarize the study and its findings. What are the take-home messages?? Please rewrite the entire conclusion part.
Reviewer 2 Report
The article in question has scientific merit, but I suggest adjustments for publication in order to increase its importance.
Cyclic heat stress has been widely used in poultry research to approach the reality of broiler farming in several countries. The main objective of this article would be to validate this form of stress in assessments mainly related to intestinal permeability, bone mineralization and meat quality. This being the objective, I suggest that the authors focus on the results found in the evaluations carried out and their possible consequences in more depth.
Material and methods:
Insert the light program used. This helps explain performance results.
Results and discussion:
Insert p and SEM values into all tables.
Remove illustrative figures. In my opinion, these figures would fit well in lectures and not in a scientific article.
I suggest putting the temperature results in a table rather than a graph form. If possible with statistical evaluation.
Results:
Line 193 - Better to use a worsening in feed conversion than a reduction.
Discussion:
Focus the discussion on the results found and seek greater depth in the discussions.
Example: the results of myopathies should be further explored.
The first paragraph of the discussion should be in the Introduction.
Deepen the discussions of the results found.
The results found in thermoneutrality are excellent to show the importance of the environment in the performance and maintenance of health in broilers. Instead of exploring these results, the authors preferred to cite possible additives to be used in heat stress.
I suggest removing the last paragraph from the discussion. Effects of heat stress must be corrected primarily by actions on environmental controls in the houses. This was never mentioned in the article. These negative effects will not be efficiently corrected with the use of additives in the feed as the substances causing these effects will continue to be produced. In addition, it is not the objective of the work to discuss the use of antibiotics or other additives.
Bibliographic references: excess of references used. Avoid citations of articles with other species as much as possible.
Reviewer 3 Report
Reviewer's comments on paper “Assessment of experimental cyclic heat stress on intestinal permeability, bone mineralization, white blood cell counts and meat quality in broiler chickens” submitted to Animals.
Dear Authors,
I read with interest your manuscript and I think that it is good quality paper suitable for publication. The working hypothesis is clearly stated, performed analysis are properly selected, described and discussed (with some minor exceptions detailed below). The final conclusions from the obtained results are drawn correctly.
Below I give some of my recommendations and suggestions for the Authors:
L85 what was the light schedule ? Was HS applied during dark hours?
L95 Recommended by who ? Any references?
L104 The description of tibia strength evaluation procedure is missing in materials and methods section.
L111-112 Detailed information about producers needed.
L135-136 ref [33] does not give any information about WB sensory evaluation
L139-140 so, left or right fillet ?
L147 shouldn’t it be -20C?
L166 t-test for discrete point ordinal scale for WB WS tactile evaluation scale and visual myopathy score rating is not a proper one. Chi-square test or contingency tables preferred.
L300 See also a recent preclinical broiler chicken-based study doi:10.3390/jcm11010205
Thanks for the opportunity to review this interesting manuscript.
Round 2
Reviewer 1 Report
A significant improvement has been made during the revision. In addition, the authors addressed the majority of my comments. I agree that the brief report is a better and more appropriate format to deliver this study than the research article.
Author Response
Thank you
Reviewer 3 Report
The article has been improved in accoreance with reviewers' suggestions.
Author Response
Thank you